# PDDL+ Models for Deployable yet Effective Traffic Signal Optimisation

**Primary Keywords:** *Applications*

## Abstract

The use of planning techniques in traffic signal optimisation has proven effective in managing unexpected traffic conditions as well as typical traffic patterns. However, significant challenges concerning the deployability of generated signal plans remain, as planning systems need to consider constraints and features of the actual real-world infrastructure on which they will be implemented. To address this challenge, we introduce a range of PDDL+ models embodying technological requirements as well as insights from domain experts. The proposed models have been extensively tested on historical data using a range of well-known search strategies and heuristics, as well as alternative encodings. Results demonstrate their competitiveness with the state of the art.

## Introduction

There is a growing interest in the use of automated planning and scheduling techniques for urban traffic control (Smith 2020), specially for traffic signal optimisation. The use of automated planning, in particular, yields the benefits of great flexibility in terms of goals that can be described and achieved, and a centralised overview of the target region. The problem of traffic signal control has been tackled using PDDL+ automated planning (Vallati et al. 2016; Antoniou et al. 2019), with knowledge models subsequently re-engineered by McCluskey and Vallati (2017) and highly effective domain-specific heuristics introduced by Percassi et al. (2023). This line of research leads to approaches that are capable of efficiently generating good quality signal plans with significant benefits in terms of congestion and emissions reduction. As with other applications of planning that are actually deployed to the real world, however, technological constraints are very specialised and may well change drastically, even within the same area, when faced with a different deployment infrastructure. This is because, in any application area, the constraints and features of the infrastructure that will implement plans need to be accounted for, and shape the capabilities and characteristics of the planning systems.

In this paper, we report on the process of adapting previous automated planning techniques for traffic signal optimisation to cope with a legacy traffic control infrastructure which is common in *urban* areas of the UK, forming the basis of Urban Traffic Control (UTC) technology. To do so, the knowledge models have to be redesigned to incorporate extra constraints and features that take into account the peculiar deployment constraints of the infrastructure.

More specifically, we introduce three new PDDL+ models which enable domain-independent planning engines to produce deployable signal plans on UTC. For the purposes of comparison and collecting real-world data, we use a region where normally the traffic reactive SCOOT (Taale, Fransen, and Dibbits 1998) control system is in operation within the UTC architecture. We extensively test the introduced models to assess their capabilities and to evaluate the impact of different language features on the performance of a wide range of domain-independent search techniques and heuristics. Finally, we show that the generated plans are comparable with the state-of-the art, and ready to be deployed in the real world.

## Research Context

In essence, traffic signal control is the problem of determining the optimal green length for each signal in a set of traffic signals, which may be dispersed around a region consisting of several spatially-close traffic junctions. The problem is structured by grouping sets of green lit signals into *stages*: each signal in a stage shares the same green time, is situated in the same junction, and collectively lets traffic flow through the junction in a safe manner. This structuring leads to the simpler problem of determining the optimal green length for each stage.

The goal of the traffic signal problem could be as general as minimising average traffic delay in the region, or as specific as alleviating the extreme delays of traffic exiting major city events and passing through the region. Typical traffic engineering mechanisms to solve this problem are traffic-reactive, and make decisions in real-time on how to change stage duration from cycle to cycle, based on sensor data (a cycle is the time taken to move through a sequence of stages of the junction, starting at a distinguished stage one, and returning back to stage one). Constraints on the problem include the legal and practical restrictions on the minimum and maximum length of each stage, as well as on the minimum and maximum length of the overall traffic signal cycle. While traffic local-reactive technology is essential (in particular, the presence of a stage within a cycle may be demand driven, so that stages are used by a single junction only

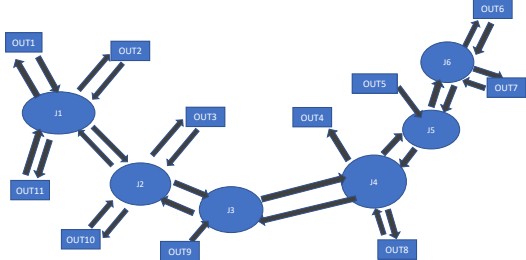

Figure 1: A simplified overview of the modelled urban region, in terms of junctions (circles), links, and boundaries (rectangles). For readability, the map is not correctly scaled.

when needed), our research has concentrated on producing plans that coordinate junctions working together for periods of time, with the aim of achieving goals given the knowledge of traffic demands in a region. E.g., the knowledge that 1,000 vehicles will attempt to leave a city centre at a particular time after a large leisure event forms part of an initial state to construct a plan alleviating the extreme delays that happen when purely local-reactive technology is used.

Although we have exported our technology to other parts of the UK, for illustration we focus on an urban region located in the Kirklees council area within West Yorkshire, United Kingdom. It encompasses a major corridor that connects the Huddersfield ring road with the M1 highway and extends to the southern part of the Kirklees council. This corridor serves as a crucial route for commuters and delivery vans travelling to Huddersfield town or moving between the M62 and M1 highways, as well as for people joining or leaving events hosted at the nearby John Smith's Stadium. Spanning approximately 1.3 kilometres, it comprises six junctions and 34 road links. Each junction consists of four to six stages, accommodating between 10 and 17 valid traffic movements. The 6 junctions sit in a single SCOOT region, i.e., an area where all of them are controlled by the SCOOT system and information between nearby junctions are shared to minimise overall delay for vehicles navigating the corridor. Figure 1 provides a simplified representation of this urban area.

SCOOT (Taale, Fransen, and Dibbits 1998) is a traffic reactive control mechanism used widely around the world, and is aimed at handling cycle-to-cycle changes in demand. In response to changes in traffic flows, SCOOT would gradually adapt and adjust the traffic signal timings of a set of managed neighbouring junctions. SCOOT is dependent on its own local data sensors, usually inductive loops embedded in the road surface, and stores sensed data and operational information in a dedicated database; for details about the extraction process to create planning knowledge from the database, the interested reader is referred to Bhatnagar et al. (2022b). Using this historical data, one can ensure that the *PDDL+ simulation* of traffic plans is as accurate as possible, since one can simulate the SCOOT plans in the PDDL+ model and check the data from the simulation against the historical records in the database.

We leverage on the architecture proposed by Bhatnagar

et al. (2023) for generating and simulating traffic plans in areas controlled by SCOOT systems, by means of PDDL+ planning. The idea of the architecture is to exploit the existing infrastructure deployed in an urban region and to use the planning-enhanced traffic signal control module as a plugin that can be activated when needed, with goals either specified by a human operator or pre-defined for routine interventions. This modular solution has a number of benefits: (i) minimising costs for traffic authorities, that can reuse what is already available; (ii) maximising robustness and safety, as generated plans are implemented and checked by an existing and extensively validated system, and (iii) supporting maintenance and continuous improvements, as the planning system can be modified and swapped with no implications for the rest of the system. It is worth noting that the models proposed in this paper are agnostic with regard to the deployment architecture, but are designed to support the constraints imposed by the underlying UTC architecture.

When moving to generated plan deployment, two main technological constraints that need to be addressed, and that emerged by recent trials on the target UTC are: (i) for each junction, the length of the stages can not be modified arbitrarily; instead, configuration of cycles (i.e., the specification of the length of every stage in the cycle) can only be selected from a predefined set, and (ii) traffic engineers involved in the trials require all the cycles to have the same duration. The reason for (i) is that configurations need to be uploaded into the UTC system at least one day in advance; the reason for (ii) is that the synchronisation between junctions needs to be maintained to avoid disrupting the signal offsets (aka green wave) along a corridor of connected links.

## Existing PDDL+ Models

A region of the urban road network is modelled as a directed graph, where *edges* represent road links and *vertices* represent junctions. One special vertex represents the external area of the modelled region. Essentially, vehicles enter (leave) the network via links connected to the external area, which represents the demand for the modelled region. Each road link has a specified maximum occupancy, indicating the maximum number of vehicles allowed on the road simultaneously, and a current occupancy, representing the current number of vehicles on the road. Traffic at junctions is regulated by *flow rates* assigned to pairs of road links, that encode the valid traffic movements. For two road links, $r_x$ and $r_y$, connected to a junction $i$, and a traffic signal stage $p$, the flow is active when $r_x$ is the incoming link, $r_y$ is the outgoing link, and the traffic signal for $i$ is green during stage $p$. Flow rates represent the number of vehicles, measured in Passenger Car Units, that can move from $r_x$, through $i$, and into $r_y$ per unit of time.

Junctions are associated with a sequence of traffic signal stages, where each stage is either fixed (it must occur) or demand driven (it will occur if the demand is there for it). In our models, we assume all stages as fixed, and allow the traffic controller to skip them if no demand is present. We use the *next* predicate to define the sequence of stages. The active traffic signal stage determines the flow rates corresponding to the green lights. For each stage, there is a specified

```
1   (:event triggerChange
2    :parameters (?p1 ?p2 - stage ?j - junction)
3    :precondition (and
4      (inter ?p2) (contains ?j ?p1) (next ?p ?p2)
5      (>= (intertime ?j) (- (interlimit ?p1) 0.1)))
6    :effect (and
7      (not (inter ?p1))
8      (active ?p2)
9      (assign (intertime ?j) 0)))
```

Figure 2: PDDL+ *triggerChange* event for transitioning from a stage $p_1$ to the next one $p_2$ over junction $j$.

```
1    (:action extendStage
2     :parameters (?p - stage ?i - junction)
3     :precondition (and
4       (controllable ?i) (contains ?i ?p)
5       (active ?p)
6       (< (+ (gt ?p) (gran)) (maxgt ?p))
7       (< (+ (ct ?i) (gran)) (maxct ?i)))
8     :effect (and
9       (increase (gt ?p) (gran))
10      (increase (ct ?i) (gran))))
```

Figure 3: PDDL+ *extendStage* action for extending the duration of a stage $p$ referring to the junction $j$.

range for the minimum and maximum stage length. A signal plan determines the length within this range. Each stage ends with an "intergreen" period, which is the time required for a signal to change to green while allowing for stacked vehicles in the middle of the junction to clear and/or providing time for pedestrian crossings. Intergreens have fixed minimum and maximum limits, and in operation vary in length depending on demands. In our current implementation, their lengths are fixed and estimated by utilising relevant historical data.

Processes are used to model the continuous flow of vehicles through a junction and to track the duration of time stages and intergreens. The limits and boundaries are managed through PDDL+ events. An important event, which will be exploited later, is the one for handling the transition from a stage to the following one on the same junction. Specifically, given two stages $p_1$ and $p_2$, such that $p_2$ follows $p_1$, and a junction $j$, the event *triggerChange*$(p_1, p_2, j)$ is triggered when $p_1$, including its intergreen time, has been completed, enabling $p_2$ on $j$. Figure 2 provides an excerpt from the model of the lifted event *triggerChange*.

In early works (McCluskey and Vallati 2017; Franco et al. 2018), the proposed knowledge model supported traffic signal optimisation through an action *switchStage*$(p, i)$, where $p$ is a stage and $i$ is a junction. This action allows for interrupting the *currently* ongoing stage and switching to the next one, provided that $p$ has reached the preset minimum duration. This model has a significant drawback: it is not possible to effectively impose constraints on the minimum and maximum length of a cycle, as required by the regulations. To overcome this issue, Percassi et al. (2023) introduced an improved version of the PDDL+ model, where stages' length can be modified with a given granularity, and cycles' length constraints can be modelled and taken into account. The main improvement of this model is the decoupling of the *switchStage* action into two different actions, namely *extendStage*$(p, i)$ and *reduceStage*$(p, i)$. Figure 3 provides an excerpt from the model of the lifted action *extendStage* for increasing the duration of a running stage. Such an action is applicable if junction $i$ is controllable, stage $p$ belongs to junction $i$, and importantly, if the increase in the duration of the greentime (gt) does not violate the overall maximum cycle length for $i$ (maxct) and the maximum length of an individual stage $p$ (maxgt). The execution of this action results in an increase, equal to the chosen granularity, of the duration of stage $p$ and the cycle to which

it belongs. The use of these two actions allows a planning engine to effectively "trade" green time between different stages in a cycle, thus supporting overall cycles' length constraints. This model also introduced a numeric fluent *counter* for each link, used to record the number of vehicles that entered the link over the plan duration, which is then exploited for defining goal conditions.

As apparent, even the most recent model based on extend and reduce actions does not support the constraints of the UTC infrastructure, since it is not possible to use only a fixed set of predefined cycles configurations, and therefore leads to traffic signal plans that can not be deployed in the real world. In the following, we will refer to the model based on extend and reduce actions as EXRE.

## Engineering PDDL+ Models for Deployability

In this section, we propose three planning models whose solutions can be deployed in the UTC infrastructure. Their common feature is that cycles configurations have to be selected from a provided pool of candidates. The three models are denoted as Cycle by Cycle (CBC), Fixed Repetition (FIRE), and Variable Repetition (VARE). Before delving into the details of the models, let us more formally define cycles configurations. Let $j$ be a junction, and let $\mathcal{S}_j = \langle s_1, ..., s_{k_j} \rangle$ be the sequence of stages associated with $j$, also referred to as a cycle. Additionally, let $\mathcal{G}_j = \{gt_1, ..., gt_{k_j}\}$ be a set of numeric variables that track the *current* duration of each stage within $j$. A *cycle configuration of $j$* is a complete assignment over $\mathcal{G}_j$. E.g., given a junction $j$ having a cycle involving 3 stages $\mathcal{S}_j = \langle s_1, s_2, s_3 \rangle$, a configuration for $j$ is $\{\langle gt_1 = 20 \rangle, \langle gt_2 = 20 \rangle, \langle gt_3 = 50 \rangle\}$, assigning 20 secs, 20 secs, and 50 secs to the duration of the three stages.

In the proposed models, each junction has a set of predefined configurations. For example, suppose that $j$ has two possible configurations, i.e., $\mathcal{C} = \{\mathcal{C}_a, \mathcal{C}_b\}$. The duration of the stages $\mathcal{S}_j$ are represented as constants, i.e., $\mathcal{G}_j = \{gt_1^a, gt_2^a, gt_3^a, gt_1^b, gt_2^b, gt_3^b\}$, where $gt_k^c$ is the fixed duration of the stage $k \in \{1, 2, 3\}$ for the configuration $c \in \{a, b\}$. A possible set of configurations can be defined by the following predefined assignments $\mathcal{C}_a = \{\langle gt_1^a = 20 \rangle, \langle gt_2^a = 20 \rangle, \langle gt_3^a = 50 \rangle\}$ and $\mathcal{C}_b = \{\langle gt_1^b = 20 \rangle, \langle gt_2^b = 50 \rangle, \langle gt_3^b = 20 \rangle\}$. Notably, the sum of the lengths of the three stages is 90 secs for both configurations, so that the offset in the corridor is kept. Essentially, configurations $\mathcal{C}_a$ and $\mathcal{C}_b$ give priority to the 3rd and 2nd stage of the junction, respectively. The plan-

```
1  (:action changeConfiguration
2   :parameters (?p - stage ?j - junction ?c1 ?c2 -
         configuration)
3   :precondition (and
4       (inter ?p) (controllable ?j) (endcycle ?j ?p)
5       (availableconf ?j ?c2) (activeconf ?j ?c1)
6       (not (activeconf ?i ?c2)))
7   :effect (and
8       (not (activeconf ?j ?c1))
9       (activeconf ?i ?c2)))
```

Figure 4: PDDL+ *changeConfiguration* action for changing the configuration of the junction $j$ from $c_1$ to $c_2$.

ning problem hence entails the selection of the configuration for $j$ to be used, and when, between $\mathcal{C}_a$ and $\mathcal{C}_b$.

## Cycle by Cycle

In CBC, flexibility is maximised, allowing the configuration of a junction to be selected in every cycle transition. This model aims at assessing whether it is worth allowing higher degrees of freedom to the planning system. In real-world deployment, however, instead of changing configuration every cycle, it is preferable to repeat the same configuration for several consecutive cycles to minimise the overhead of the signal plan change, hence increasing robustness.

The flexible behaviour is achieved by the action *changeConfiguration*$(p, j, c_1, c_2)$, where $p$ denotes a stage, $j$ represents a junction, and $c_1$ and $c_2$ are two distinct configurations for $j$. In this context, $c_1$ denotes the currently active configuration on $j$, while $c_2$ represents the new configuration that will be adopted by $j$. To prevent redundant decision points, which would not enhance the expressive power of the model, the configuration selection can only occur at the end of the cycle, during the intergreen of the cycle's final stage. Figure 4 shows the lifted action *changeConfiguration*.

Similarly to the *extendStage* action, the action *changeConfiguration* can be applied if the junction $j$ is controllable. Additionally, the action requires that the stage $p$ is currently in the intergreen phase (predicate (inter ?p)) and $p$ is the last stage for the cycle referring to junction $j$ (predicate (endCycle ?i ?p)). This condition ensures that the configuration selection action can only be executed during the intergreen phase of the last stage of the cycle.

For the transition from $c_1$ to $c_2$, it is necessary that $c_1$ is currently active for junction $j$ (predicate (activeconf ?j ?c1)) and that configuration $c_2$ is available for junction $j$ (static predicate (availableconf ?j ?c2)). To ensure that the action is executed only when $c_1$ and $c_2$ are different, it is also required that configuration $c_2$ is not currently in execution, thus avoiding the generation of ineffective actions. The execution of this action deactivates (activates) configuration $c_1$ ($c_2$). Consequently, all events employed to manage the execution of the cycle will be conditioned to use the stage durations prescribed by $c_2$.

### Fixed Repetition

The FIRE model enforces the retaining of the selected configuration for a minimum of $k$ cycles. Once the minimum number of cycles has been reached, there is the option to change the configuration for the considered junction. This model allows a number of decision points lower than CBC. Our conjecture tested with an experimental study is that this can lead to better performance.

To track the number of completed cycles associated with the current configuration for each junction, we introduce variable (countcycle ?j), and event *trigger-change*$(p_1, p_2, j)$, which models stage transitions from $p_1$ to $p_2$ for $j$. When *trigger-change*$(p_1, p_2, j)$ triggers and $p_1$ is the last stage of the cycle, the cycle counter is increased by one. This state-dependent effect is implemented through the following conditional effect added to the effects of the *trigger-change* event:

```
1  (when
2       (endcycle ?j ?p1)
3       (increase (countcycle ?i) 1))
```

Here, the static predicate (endcycle ?j ?p1) is used to verify if $p_1$ corresponds to the last stage of the cycle for $j$.

The FIRE model adopts the same action *changeConfiguration* as the CBC one, with slight modifications. In addition to the original preconditions, it is required that the minimum number of cycles (cyclelim) has been reached; this is done by adding the precondition (>= (countcycle ?i) (cyclelim)). Also, the use of the action causes the reset of the counter, modelled by the additional effect (assign (countcycle ?i) 0).

### Variable Repetition

VARE takes control to a deeper level by allowing decisions on how many times a selected configuration has to be repeated for a specific junction. Given a junction $j$, we denote by $k_j$ the minimum number of repetitions of a configuration in $j$. Such a value can vary within a defined range, namely $\{k_{\min}, ..., k_{\max}\}$. Notably, when $k_{\min} = 1$, VARE's control capability is equivalent to that of CBC. On the one hand, it introduces additional decision points related to the choice of $k_j$ w.r.t. FIRE; on the other hand, the number of decision points related to the change of configurations can be lower, as $k_{\max}$ can be higher than the value of (cyclelim) for FIRE. We experimentally evaluated if this modelling pays off in terms of the achieved performance.

$k_j$ is modelled in PDDL+ as the numeric variable (varlimit ?j), serving a similar role as (cyclelim) in the action for changing the configuration in FIRE. However, since $k_j$ is linked to a specific junction $j$, the corresponding numeric variable is also parameterised with the object ?j representing the junction. The admissible values for $k_j$ are modelled as constant numeric variables (conflim ?l), where ?l is an object of type *limit* associated with each possible value in the range. VARE allows setting the value of $k_j$ whenever the action for changing a configuration is executed. Specifically, the action *changeConfiguration* activates an additional action, *changeLimit*$(p, j, l)$, where $p$ is a stage, $j$ is a junction, and $l$ is a limit, which assigns to (varlimit ?j) the value (conflim ?l).

Once the value of $k_j$ has been set, the remaining part of the model for managing the duration of the stages and the cycle count remains unchanged w.r.t. the model FIRE.

### Reformulation and Optimisation

The proposed models make use of PDDL features that are known as potentially problematic for PDDL+ planners, namely conditional effects and numeric assignments. Because of that, we designed reformulations of the models where such features are compiled away. In the experimental analysis, presented in the next section, we also take the occasion to empirically assess whether removing these features could result in models that are more manageable for state-of-the-art planning systems.

The introduced models allow us to take decisions *during* the last intergreen phase of a given cycle (called Intergreen Time-window). In our experiments, we also tested a variant of these models for which deciding a cycle can be done exactly *at the end* of the last intergreen phase (called Instantaneous Time-window). Given the discretisation of the time utilised for our experiments, the number of decision points for Instantaneous Time-window is usually 5 times lower than for Intergreen Time-window. Therefore, in our experiments, we also study if and how much this variant pays off. All the models and variants are available at: https://anonymous.4open.science/r/utc-models-deployable-74BD/.

## Empirical Evaluation

The experimental analysis aims to assess the capabilities and performance of the proposed models and considered language features, and to understand the ability of the resulting system to generate effective strategy plans for real-world scenarios. For this reason, the evaluation consists of two parts. First, we perform a comparison across the proposed models and formulations, using a broad range of search configurations provided by the PDDL+ planner ENHSP version 20 (Scala et al. 2020a). This planner, in addition to providing the possibility of performing customised searches in PDDL+, has proven to be very effective in dealing with traffic control problems modelled in PDDL+ (Bhatnagar et al. 2023; Percassi et al. 2023). This first extensive analysis aims to identify the best candidate, understood as a model combined with a search configuration, to be compared with state-of-the-art signal plans in the second part of the experimental evaluation. To compare against the state of the art, we consider the historical signal plans implemented by the SCOOT system, and the signal plans generated by a domain-specific heuristic working on the ExRe model (Percassi et al. 2023).

### Experimental Settings

We use an extended version of the benchmark used by Percassi et al. (2023), focusing on the corridor presented in Figure 1. We consider six scenarios in two distinct days: the 26th (referred to as day *A*), which is a Wednesday, and the 30th (referred to as day *B*), a Sunday, both in January 2022. It is important to highlight that COVID-19 restrictions were no longer in effect during that period in the United Kingdom. Each day was examined at three different time slots: the morning peak hour at 8:30 am (*morn*), noon at 12:30 pm (*noon*), and the evening peak hour at 4:30 pm (*eve*). This variation aimed to assess diverse traffic volumes and conditions. The notation used for the scenarios is expressed as *day-slot*, e.g., *A-morn*. Further, we include a scenario (*Concert*) involving exceptional traffic circumstances, pertaining to a concert held at John Smith's Stadium on Tuesday the 20th of June 2023, which attracted an approximate audience of $30,000$ people. The time considered is 4:00 pm, which is before the start of the concert. This timing is challenging for the considered corridor because there is a clash between commuters leaving the town and spectators arriving at the concert, creating two opposed traffic demands.

For each scenario, we generate multiple UTC planning problems, progressively expanding the set of explicitly considered corridor links in the goal. Specifically, for a given scenario, we create five UTC planning problems denoted as $\Pi^i_{\mathrm{UTC}}$, where $i \in \{1, ..., 5\}$. In $\Pi^i_{\mathrm{UTC}}$, the goal is represented as a conjunction involving the first $i$ links of the corridor, i.e., $G = \bigwedge_{l \in \{l_1, ..., l_i\}} \langle counter_l \geq q_l \rangle$. To illustrate, if $i = 2$, then $G = \langle counter_{l_1} \geq q_{l_1} \rangle \wedge \langle counter_{l_2} \geq q_{l_2} \rangle$, signifying that to solve $\Pi^2_{\mathrm{UTC}}$, a state must be reached wherein at least $q_{l_1}$ and $q_{l_2}$ vehicles have traversed links $l_1$ and $l_2$, respectively. This strategy demonstrated the ability to support different kinds of goals for the network, and different behaviours of planning systems: for example, a single goal link at the end of the corridor can lead to signal plans that focus on "flushing" vehicles already in the network towards the goal link, while multiple goal links require also reasoning in terms of congestion and ability of vehicles to move through different junctions. Here, we consider a uniform $q$ value of 350.

All plans generated by the new models have been validated against the ExRe model, to confirm compliance with existing requirements, and simulated on historical data via the architecture designed by Bhatnagar et al. (2022a), to assess deployability and ability to model traffic evolution.

Experiments were run on a machine equipped with Intel Xeon Gold 6140M CPU with 2.30 GHz, 8 GBs of RAM.

**Cycle Configuration Distillation** The proposed models require a set of cycle configurations. To explore how different configurations may impact the models' ability to generate effective signal plans, we have outlined three methodologies for their distillation.

The first methodology involves generating configurations synthetically. Let $j$ be a junction, and $S_j$ be the sequence of stages associated with it. The maximum-1 strategy, denoted as MAX1, generates a number of configurations equal to $|S_j|$, which in our region is at most 6. Specifically, each configuration prioritises one stage over the others while preserving a fixed duration of the cycle. In this methodology, the prioritisation is *flat*: the maximised stage has a higher duration, while the remaining stages have an equally short green time allocated.

The other two methodologies, instead, draw from the configurations historically adopted by SCOOT in the considered region. S-HIST generates configurations by looking at configurations implemented by SCOOT in the previous year at the same day of the week and time of the day. From these configurations, six are then selected following the idea of the MAX1 above, i.e., maximising different stages' lengths at a time. G-HIST employs a similar approach but considers as

potential candidates all the configurations implemented by SCOOT in the entire past year, without restrictions in terms of days and times. These two approaches aim to provide the planning approach with configurations that have been useful in the past, hence aligned with the needs of the region.

Following the instructions of traffic engineers from the local traffic authority, we use configurations with a total length of 90 seconds, and we keep the same 90 seconds value for all junctions to support the synchronisation of flows and green waves implicitly.

### Comparison of PDDL+ Models

In the PDDL+ intra-model experimental analysis, we perform a detailed comparison among the novel models, along with their different formulations, employing various search strategies and different heuristics. Given a model $\mathcal{M} \in \{\text{CBC}, \text{FIRE}, \text{VARE}\}$, we denote by $\mathcal{M}$ the base model and with $\mathcal{M}(\text{-}f)$, where $f \in \{ce, asgn\}$ (conditional effects and numeric assignments, respectively), the formulation of $\mathcal{M}$ in which the language feature $f$ has been removed.

For the FIRE model, we chose to keep the configuration for a number $k$ of cycles equal to 4 as it represents a good trade-off between stability and flexibility. This way, once a configuration is chosen, it is maintained for 6 minutes in the real world, and at most 3 configurations are needed for a 15-minute strategy. As for VARE, we considered the range of $k \in \{4, \ldots, 10\}$, where the minimum value is the same as the one chosen for FIRE, and the maximum corresponds to a real-world duration of 15 minutes. Values beyond this range are not useful, as simulations after 15 minutes diverge from the real-world behaviour due to the shifting of underlying turnrates factors (Bhatnagar et al. 2022b).

For each model in this analysis, we show the results of the optimised variant in which there is a single cycle decision point (Instantaneous Time-window). This had a strongly beneficial impact on performance, when compared to the Intergreen Time-window variant. A comparison is shown in Figure 5 (left) and discussed later in this section.

The considered search strategies are greedy best-first search (GBFS) and A* (Hart, Nilsson, and Raphael 1968), and the adopted heuristics included $h^{add}$, $h^{max}$, and $h^{mrp}$ (Scala et al. 2020b). A *search configuration* is defined as the combination of a search strategy along with a particular heuristic. Given the large number of systems obtainable (7 models and 6 search configurations), we rank their performance using the IPC quality score, calculated based on the makespan.[1] Makespan provides an idea of the effectiveness of the implemented signal plans in quickly reaching goals.

Out of all the considered *search configurations*, the best performing across all models is GBFS with $h^{add}$, except for the FIRE(-*asgn*) model, where the best configuration is GBFS with $h^{rmp}$. In the following, we focus on results achieved using such search configurations, as they represent the top performance that each model can deliver in the considered settings.

Table 1 provides scenario-by-scenario makespan results for all models. Results are aggregated across scenarios ac-

---

[1] https://ipc2023-classical.github.io/ for details on IPC Score.

| Scenario | Cycle | CBC | | FIRE | | VARE | | |
| --- | --- | --- | --- | --- | --- | --- | --- | --- |
| | | B | *-asgn* | B | *-ce* | B | *-asgn* | *-ce* |
| *A-morn* | MAX1 | 3.0 | 3.1 | 3.2 | 3.2 | 2.8 | 2.8 | 2.7 |
| | S-HIST | **4.9** | **4.9** | **4.9** | **4.9** | **4.9** | **4.9** | **4.9** |
| | G-HIST | **4.9** | **4.9** | **4.9** | **4.9** | **4.9** | 3.9 | **4.9** |
| *A-noon* | MAX1 | 2.9 | 2.9 | 2.9 | 2.9 | 2.9 | 2.8 | 2.9 |
| | S-HIST | **4.9** | **4.9** | **4.9** | **4.9** | **4.9** | **4.9** | **4.9** |
| | G-HIST | 4.8 | **4.9** | **4.9** | **4.9** | **4.9** | **4.9** | **4.9** |
| *A-eve* | MAX1 | 3.0 | 3.0 | 3.0 | 2.8 | 3.0 | 2.4 | 3.0 |
| | S-HIST | **4.8** | **4.8** | 4.7 | 4.7 | **4.8** | **4.8** | **4.8** |
| | G-HIST | 4.7 | 4.6 | **4.8** | 4.7 | **4.8** | **4.8** | 4.7 |
| *B-morn* | MAX1 | **5.0** | **5.0** | **5.0** | 4.9 | 4.8 | 4.9 | 4.9 |
| | S-HIST | 1.0 | 4.9 | 4.9 | 4.9 | 2.9 | 2.9 | 1.0 |
| | G-HIST | **5.0** | **5.0** | **5.0** | 4.9 | 3.9 | 3.9 | **5.0** |
| *B-noon* | MAX1 | 2.8 | 2.8 | 2.8 | 2.8 | 2.8 | 2.8 | 2.9 |
| | S-HIST | **5.0** | 4.9 | **5.0** | 4.9 | **5.0** | **5.0** | **5.0** |
| | G-HIST | **5.0** | 4.9 | **5.0** | 4.9 | **5.0** | **5.0** | **5.0** |
| *B-eve* | MAX1 | 3.5 | 3.3 | 3.5 | 3.6 | 3.1 | 3.2 | 3.1 |
| | S-HIST | **5.0** | **5.0** | **5.0** | **5.0** | **5.0** | **5.0** | **5.0** |
| | G-HIST | **5.0** | **5.0** | **5.0** | **5.0** | **5.0** | **5.0** | **5.0** |
| *Concert* | MAX1 | 3.0 | 2.8 | 2.9 | 3.1 | 2.7 | 2.8 | 2.7 |
| | S-HIST | **4.9** | 4.7 | 4.6 | 4.6 | 4.5 | 4.6 | 4.6 |
| | G-HIST | 4.6 | 4.8 | 4.6 | 4.8 | 4.6 | 4.6 | 4.6 |
| Σ | MAX1 | 23.2 | 22.8 | **23.3** | 23.1 | 22.1 | 21.6 | 22.2 |
| | S-HIST | 30.4 | **34.0** | **34.0** | 33.9 | 32.1 | 32.1 | 30.2 |
| | G-HIST | 33.9 | 34.1 | 34.0 | **34.2** | 33.1 | 32.1 | 34.1 |
| Σ | Σ | 87.6 | 90.9 | **91.3** | 91.2 | 87.3 | 85.8 | 86.4 |

Table 1: IPC-Score results for the makespan across the models and their different formulations (B stands for the base model). The results are split according to the cycle configuration strategy adopted. Best results are in bold.

cording to the exploited cycle configurations, offering preliminary insights into the impact of injected configurations on different models. In terms of makespan, all the models tend to provide similar results across the scenarios, with significant variations emerging when different cycle configurations are used. Unsurprisingly, MAX1 is the distillation approach that leads to the worst results, while S-HIST and G-HIST allow to achieve plans of very similar quality. Turning our attention to the models, the top-performing one is FIRE in both of its formulations, followed by CBC in its version without numeric assignments, i.e., FIRE(-*asgn*). It is interesting to note that the higher flexibility provided by the CBC formulation is not reflected in better performance, while the tradeoff between flexibility and complexity provided by FIRE seems to better support the generation of good quality signal plans. Further, removing the use of assignments in CBC leads to better results but only in a single scenario, namely *B-morn*, as it increases coverage.

Table 2 sheds some light on the results by also showing coverage (number of solved instances) and IPC-Score for expanded nodes and planning time. These results provide additional insights into Table 1. It is evident that the use of FIRE-based models achieves the highest coverage, successfully solving all considered instances. This outcome is

| | CBC | | FIRE | | VARE | | |
|---|---|---|---|---|---|---|---|
| | B | *-asgn* | B | *-ce* | B | *-asgn* | *-ce* |
| *Coverage* | | | | | | | |
| MAX1 (35) | **35** | **35** | **35** | **35** | **35** | 34 | **35** |
| S-HIST (35) | 31 | **35** | **35** | **35** | 33 | 33 | 31 |
| G-HIST (35) | **35** | **35** | **35** | **35** | 34 | 33 | **35** |
| Σ (115) | 101 | **105** | **105** | **105** | 102 | 100 | 101 |
| *Score(ExpNodes)* | | | | | | | |
| MAX1 | 9.7 | 9.9 | **17.4** | 16.7 | 4.6 | 4.7 | 4.8 |
| S-HIST | 7.0 | 11.1 | **16.9** | 16.1 | 8.6 | 8.2 | 8.0 |
| G-HIST | 8.1 | 9.7 | **17.9** | 17.1 | 8.3 | 7.3 | 7.1 |
| Σ | 24.7 | 30.6 | **52.1** | 49.9 | 21.6 | 20.2 | 19.9 |
| *Score(PTime)* | | | | | | | |
| MAX1 | 25.4 | 24.5 | 26.5 | **27.0** | 18.4 | 13.8 | 18.7 |
| S-HIST | 19.1 | 22.1 | **24.2** | 22.8 | 19.5 | 13.8 | 18.7 |
| G-HIST | 20.7 | 21.5 | **25.1** | 24.6 | 19.9 | 13.6 | 19.8 |
| Σ | 65.2 | 68.1 | **75.8** | 74.4 | 57.8 | 41.2 | 57.2 |

Table 2: Coverage and IPC-Score about expanded Nodes (*ExpNodes*) and planning time (*Ptime*) across the models and their different formulations (B stands for the base model). The results are split according to the cycle configuration strategy adopted. The best results are in bold.

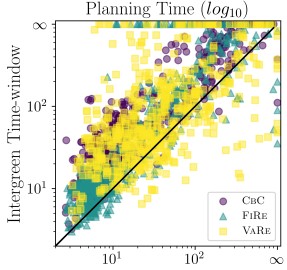
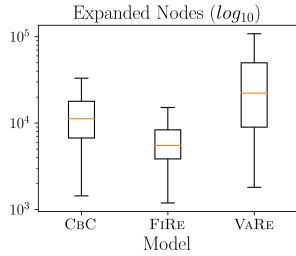

Figure 5: (*Left*) Planning times obtained by comparing the models using Instantaneous Time-windows, compared to the models that do not employ this optimisation. (*Right*) Whisker plot of expanded nodes across different models.

also reflected in the makespan score, positioning FIRE as the most promising model among those evaluated. Indeed, many of the makespan score variations from Table 1 can largely be attributed to discrepancies in coverage. For instance, in the CBC and VARE(*-ce*) models, four problems remain unsolved in *B-morn* (compare with the *B-morn* S-HIST entry in Table 1). This scenario is characterised by a low influx of vehicles in the network, causing the plan solutions to be much longer compared to other instances with a higher number of vehicles. As a consequence, models characterised by greater flexibility, and thus more degree of freedom, suffer more in terms of search effort.

Results in Table 2 also show that FIRE models allow domain-independent planning systems to solve problems quickly, compared to the other variants, with also a smaller number of nodes expanded during the search. It is also worth highlighting that for CBC (VARE) the use of assignments in the code has a slight detrimental (incremental) effect on performance. This is somehow surprising as it is generally known that assignments lower the performance for PDDL2.1 models; in VARE, this can be because removing assignments between different numeric variables leads to a larger number of ground actions in the encodings. These results offer a different perspective on the impact of that specific PDDL feature in the context of challenging PDDL+ applications. Surprisingly, the use of conditional effects is beneficial for performance. Both FIRE and VARE allow to deliver best performance when conditional effects are in use.

Turning our attention to the importance of the time-window optimisation, Figure 5 (left) compares the planning time achieved when using the three proposed models with all possible reformulations, with and without the time window optimisation. Such optimisation brings an outstanding benefit in terms of planning time (points above diagonal are those where the optimisation has a beneficial impact). The VARE model is the only one where there are some cases where the optimisation reduces performance, but the overall positive impact is also true for this class of models. The right part of Figure 5 provides a whisker plot comparing the number of expanded nodes across the three models. The obtained results confirm that the varying levels of freedom in the models have repercussions in terms of the search effort.

Taking a closer look at the generated plans, we investigate if they include cycle configuration changes, or whether solutions are generated by keeping the starting configuration on all junctions. In all cases, the solution plans include changes to the configurations, suggesting that despite exploiting domain-independent approaches, the models allow the planning engine to reason to improve traffic conditions. More specifically, the CBC model generates a much higher number of configuration changes w.r.t. FIRE and VARE. The maximum number of changes encountered in a plan for the three models is 98, 33, and 26, respectively. This is somehow expected, as CBC has the greatest degree of freedom.

FIRE is the most promising model, producing plans with less computational effort due to its good tradeoff between flexibility and effectiveness, and is used in the next section.

## Comparison Against the State-of-the-art

We are now in the position to compare the plans generated by FIRE with the plans historically implemented by SCOOT in the reference region. Additionally, we consider plans obtained by a domain-specific planning approach designed for the EXRE model, that utilises a domain-specific heuristic, $h^{\text{TSO}}$, combined with GBFS (Percassi et al. 2023).

All traffic signal plans are evaluated in simulation, utilising the simulation environment introduced in (Bhatnagar et al. 2022a). For the comparison, to provide a well-rounded performance overview, we rely on the metrics proposed by Percassi et al. (2023): $0 \leq \mu_Z(occ_{\mathcal{C}}) \leq 1$ represents the average occupancy, normalised in relation to the maximum capacity of the links in the west-to-east corridor direction; a value close to one indicates a high level of congestion.

| Scenario | Approach | $\mu_Z$ | $count_{\mathcal{C}}$ | $in$ | $middle$ | $out$ |
|---|---|---|---|---|---|---|
| *A-morn* | Max1 | 0.22 | 738.4 | 326.8 | 163.0 | 159.6 |
| | S-Hist | 0.17 | 1088.7 | **417.1** | 248.9 | 221.6 |
| | G-Hist | 0.16 | 1085.6 | **417.1** | 248.4 | 224.9 |
| | $h^{\text{TSO}}$ | **0.13** | **1108.8** | **417.1** | **253.6** | **235.8** |
| | $\mathcal{H}$ | 0.28 | 887.4 | **417.1** | 221.0 | 181.7 |
| *A-noon* | Max1 | 0.32 | 814.0 | 415.5 | 160.4 | 181.3 |
| | S-Hist | 0.31 | 1225.3 | **551.5** | 244.7 | 249.4 |
| | G-Hist | 0.3 | 1212.6 | **551.5** | 243.8 | 250.3 |
| | $h^{\text{TSO}}$ | **0.16** | **1268.4** | 547.6 | 261.2 | **264.8** |
| | $\mathcal{H}$ | 0.35 | 1138.3 | **551.5** | 270.9 | 227.3 |
| *A-eve* | Max1 | 0.43 | 833.5 | 526.2 | 166.4 | 197.4 |
| | S-Hist | 0.39 | 1204.3 | **614.7** | 245.1 | 258.0 |
| | G-Hist | 0.39 | 1209.0 | **614.7** | 245.1 | 260.3 |
| | $h^{\text{TSO}}$ | **0.15** | **1437.0** | 599.9 | 298.5 | **292.8** |
| | $\mathcal{H}$ | 0.4 | 1317.9 | **614.7** | 309.1 | 271.9 |
| *B-morn* | Max1 | 0.04 | 454.1 | **173.8** | 103.0 | 94.7 |
| | S-Hist | **0.02** | 454.4 | **173.8** | 104.7 | 94.8 |
| | G-Hist | 0.03 | 463.8 | **173.8** | 105.1 | 93.1 |
| | $h^{\text{TSO}}$ | **0.02** | **468.3** | **173.8** | 108.6 | **97.2** |
| | $\mathcal{H}$ | 0.07 | 417.8 | **173.8** | 102.6 | 83.2 |
| *B-noon* | Max1 | 0.34 | 768.8 | 400.4 | 161.1 | 169.0 |
| | S-Hist | 0.34 | 1220.2 | **560.6** | 246.9 | 243.8 |
| | G-Hist | 0.34 | 1214.7 | **560.6** | 246.9 | 238.6 |
| | $h^{\text{TSO}}$ | **0.17** | **1322.0** | 557.7 | **279.4** | **259.0** |
| | $\mathcal{H}$ | 0.73 | 612.9 | 558.9 | 146.1 | 74.9 |
| *B-eve* | Max1 | 0.16 | 678.2 | 283.5 | 148.6 | 163.7 |
| | S-Hist | **0.1** | 944.3 | **353.4** | 206.2 | 207.6 |
| | G-Hist | **0.1** | **944.4** | **353.4** | 206.2 | **209.2** |
| | $h^{\text{TSO}}$ | **0.1** | 943.9 | **353.4** | **208.2** | 207.0 |
| | $\mathcal{H}$ | 0.48 | 606.3 | **353.4** | 166.6 | 89.8 |
| *Concert* | Max1 | 0.52 | 1163.5 | **612.8** | 176.1 | 351.5 |
| | S-Hist | 0.74 | 1356.9 | **612.8** | 228.9 | 357.9 |
| | G-Hist | 0.73 | 1408.5 | **612.8** | 244.3 | 344.9 |
| | $h^{\text{TSO}}$ | 0.64 | 1492.8 | **612.8** | 269.2 | 386.6 |
| | $h^{\text{TSO}}_{\star}$ | **0.45** | **1628.2** | **612.8** | **308.7** | **409.7** |

Table 3: Comparison between the best planning system obtained, i.e., GBFS+$h^{max}$ applied to FiRe for different kinds of cycle configurations, and the state-of-the-art results, i.e., GBFS+$h^{\text{TSO}}$ applied to ExRe, and the historical strategy implemented by Scoot ($\mathcal{H}$) or $h^{\text{TSO}}_{\star}$. Best results are in bold.

$count_{\mathcal{C}}$ is the total number of vehicles that have moved in the corridor during the simulation. $in/mid/out$ is the total number of vehicles that have entered from the western entry points, crossed the middle of the corridor (between $J3$ and $J4$), and exited from the eastern exit points, respectively.

Table 3 (top) presents the results of the comparison for days *A* and *B*. Each sub-table, corresponding to a specific scenario, displays the metrics obtained for the FiRe model tested with different stage configurations, $h^{\text{TSO}}$, and the historical data generated by Scoot (denoted by $\mathcal{H}$). For the FiRe-based and $h^{\text{TSO}}$ models, the results are reported for the problem that maximised the metric $count_{\mathcal{C}}$.

The use of cycle configurations derived from historical data (S-Hist and G-Hist) enhances the results obtained from the FiRe model also in terms of traffic-related metrics. Specifically, the combination of FiRe and S-Hist (G-Hist) yields a value of $count_{\mathcal{C}}$ slightly lower than that

recorded in historical data on day *A*. Overall, the counter for this combination is better than for $\mathcal{H}$ in 5 out of 6 instances. Simultaneously, the counter is marginally lower than that obtained by $h^{\text{TSO}}$ in 5 out of 6 instances. It is worth reminding that the comparison with $h^{\text{TSO}}$ is biased in favour of $h^{\text{TSO}}$, since it relies on the ExRe model, which provides a degree of freedom that is beyond the capabilities of FiRe, and which, differently from FiRe, is driven by a domain-specific heuristic. More importantly, $h^{\text{TSO}}$ leads to signal plans that can not be deployed in the region due to the technological constraints of the UTC infrastructure. Another observation is that FiRe consistently generates plans that reduce corridor congestion w.r.t. $\mathcal{H}$, albeit to a lesser extent compared to $h^{\text{TSO}}$. An example is the *B-noon* scenario, where the number of moved vehicles is roughly double while halving the overall congestion level (*middle* and *out* values are much higher).

Table 3 (bottom) focuses on the *Concert* scenario, where significant opposed traffic flows navigate the area. This scenario differs significantly from the previous ones. Firstly, it involves exceptional traffic conditions, and secondly, historical data where Scoot is in operation is not available. This is because the strategy implemented in the real-world on that occasion was generated by leveraging a plan produced by $h^{\text{TSO}}$ and then manually modified by traffic engineers, according to their knowledge, to make it deployable on the Scoot infrastructure. This variant of $h^{\text{TSO}}$ is denoted as $h^{\text{TSO}}_{\star}$, and should be regarded as the best possible performance achievable by merging human experience and planning capabilities. Unsurprisingly, $h^{\text{TSO}}_{\star}$ delivers the best overall performance. FiRe with G-Hist achieves slightly lower results than $h^{\text{TSO}}$ in terms of vehicles moved through the corridor, and interestingly, the Max1-based variant attains lower congestion levels, but this appears to be because it creates a bottleneck at the start of the corridor (*middle* and *out* values are very low). All the approaches allow an equal number of vehicles to enter from the West entry point ($in$), but for FiRe and $h^{\text{TSO}}$ *middle* and *out* are lower than for $h^{\text{TSO}}_{\star}$; this is because the implemented plans –being generated in advance– include all stages of all cycles, while the Scoot system that operates in real-time can skip optional (demand-only) stages, for instance pedestrian crossings or cross-flow traffic, if there is no demand.

Overall, the introduced FiRe model allows a domain-independent planning engine to deliver plans that are comparable with the state of the art and that, differently from the state of the art, can be continuously deployed.

## Conclusions

In this paper, we demonstrated how we adapted planning models to generate deployable traffic signal plans for a specific UTC infrastructure, widely used in the UK. In the process, we designed three new PDDL+ models that support the use of domain-independent planning engines for the task. The large experimental analysis, performed using real-world data, demonstrated the capabilities of the models and shed some light on the impact of some advanced language features. Future work will focus on investigating more sophisticated techniques for cycle configurations distillation, and on extending to additional urban regions.

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
