# OpenReview forum: "PDDL+ Models for Deployable yet Effective Traffic Signal Optimisation"
_icaps-conference.org/ICAPS/2024/Conference — ICAPS 2024_

### Official Review · Reviewer_hSRh · 2024-01-21

**Significance And Importance:** 2
**Soundness:** 3
**Novelty:** 2
**Clarity:** 3
**Overall Evaluation:** 1
**Confidence:** 5

**Weaknesses:**

1: Minor weaknesses that are easily fixable.

**Contributions Of The Paper:**

This research addresses the application of automated planning for optimizing urban traffic signal control, specifically within the legacy infrastructure of the UK's Urban Traffic Control (UTC) technology. Three new PDDL+ models are introduced to adapt automated planning techniques for UTC, considering constraints like predefined stage configurations and synchronized cycle durations. The study evaluates the performance of these models in a region using the SCOOT traffic control system.

The main contribution of the paper is the adaptation of planning to legacy UTM infrastructure, and the exploration of the proposed models and method in a real world setting.

**Ethical Considerations:**

(1) Not Applicable: The paper does not have any ethical considerations to address

**Nomination For Best Paper:**

No

**Questions For Authors:**

No futher questions beyond the previous comments.

**Reproducibility:**

3: Authors describe the implementation and domains in sufficient detail.

**Strengths Of The Paper:**

The paper is clearly written and easy to follow,  the ideas presented in it are explained and in a satisfactory manner. The practical relevance of the paper is excellent,  and the empirical results and exploration of the models, satisfies the requirements from an application paper.

**Weaknesses Of The Paper:**

I would liked to see in the paper a simple discussion on the computation complexity induced by each of the models.
Also the paper is positioned in the research literature on a very narrow line of papers, it would be beneficial to position it better in the traffic models literature, and explaining why the approach in it is preferred.

---

> ### Author Rebuttal · Authors · 2024-01-26
>
> We thank the reviewer for the supportive feedback.
>
> In terms of complexity induced by the models, CbC and FIRE are similar, as the actions allow to choose for each junction a configuration to apply -- with the notable difference that in CbC a decision can be taken at every cycle, while in VARE only every k cycles. In the case of VARE, the complexity increases as, on top of the above, there is also the additional decision of the minimum number of repetitions. We will add a further discussion around induced computational complexity in the paper.
>
> We agree that the related work analysis in the paper is limited; this is due to the limited available space. In the CRC we will buy an additional page, as allowed by AAAI, and expand the related work discussion, as well as the other points raised in this review and in the others.

---

### Official Review · Reviewer_dPRe · 2024-01-22

**Significance And Importance:** 2
**Soundness:** 4
**Novelty:** 2
**Clarity:** 4
**Overall Evaluation:** 2
**Confidence:** 4

**Weaknesses:**

1: Minor weaknesses that are easily fixable.

**Contributions Of The Paper:**

The main contribution of the paper is an analysis of different PDDL+ planning-based approaches to Urban Traffic Control for a specific region in the UK. The problem concerns determining optimal durations of green light stages in a set of traffic signals spread across several nearby junctions. The paper introduces three distinct variations of the UTC PDDL+ model: CbC, FiRe, and VaRe. The authors conduct a thorough evaluation of the differences between the PDDL+ model variants with respect to performance on historical data, as well as runtime, coverage, and expanded nodes. Further, the best performing model configurations are then compared against state-of-the-art approaches and the currently used SCOOT framework.

**Ethical Considerations:**

(1) Not Applicable: The paper does not have any ethical considerations to address

**Nomination For Best Paper:**

No

**Questions For Authors:**

1. When compiling away the conditional effects and numeric assignments, does that mean that they are replaced with a different but semantically equivalent implementation, or that they are completely removed, resulting in a more abstracted/simplified version of the UTC PDDL+ model?
2. Have you verified with a UTC subject matter expert how realistic is each PDDL+ model variant and the selected parameters (e.g., number of cycles k)?
3. Why does the time window optimization reduce performance in the VaRe model in some cases?

**Reproducibility:**

2: Some details are missing, but the paper still appears to be replicable with some effort.

**Strengths Of The Paper:**

The paper describes a very important and interesting real world application of PDDL+ planning for Urban Traffic Control. The authors clearly explain the UTC-related concepts to better justify the use of PDDL+ and AI Planning in general. The paper does well in discussing the the different selected PDDL+ model variants in detail and provides reasoning behind each variant. The entire experimental evaluation is well thought-out. The analysis is thorough and addresses all obvious questions about the presented results and findings. The comparison with SOTA is also comprehensive and provides a good discussion on the current state of SOTA an how the presented PDDL+ models could improve on them when deployed in the real world. Overall, this is a very good application paper.

**Weaknesses Of The Paper:**

The authors should better emphasize the differences in the PDDL+ models to distinguish from Percassi et al. (2023).

Minor comments:
- figure 4: ?i should be ?j in some predicates (same in line 304 (endCycle ?i ?p) ).

---

> ### Author Rebuttal · Authors · 2024-01-26
>
> We thank the reviewer for the supportive feedback, and for spotting the typo in using ?i and ?j. In the CRC, we will further emphasise the differences between the proposed models and the existing ExRe proposed in Percassi et al.
>
> Q1. The compilations preserve the semantics of the implementations, hence the models are comparable. We will clarify this in the paper and add an explanation in the final repository.
>
> Q2. The technological constraints in new models were requested by traffic engineers from the local traffic authority. Also, parameters for the models and for the experimental analysis, have been defined together with traffic engineers.
>
> Q3. This is an excellent question. We investigated it but we did not find a definitive answer. Our speculation is that, due to the complex dynamics of the modelled domain, there are cases where assessing the conditions before the end of the cycle may be advantageous for the domain-independent heuristics.

---

### Official Review · Reviewer_kQMv · 2024-01-22

**Significance And Importance:** 1
**Soundness:** 3
**Novelty:** 2
**Clarity:** 4
**Confidence:** 3

**Weaknesses:**

-1: Major weaknesses requiring significant work to be addressed for the paper to be accepted.

**Contributions Of The Paper:**

Paper 146 looks at PDDL+ solving for urban traffic control. Different planners were applied to solve the PDDL+ problems, which in general are undecidable. The main contributions seem to be the engineering of PDDL+ models approaches were called Cycle by Cycle (CbC) Fixed Repetition (Fire), and VariableRepition (Vare), together with some (manual) reformulation and optimization like instantaneous time windows.

**Ethical Considerations:**

(1) Not Applicable: The paper does not have any ethical considerations to address

**Nomination For Best Paper:**

No

**Overall Evaluation:**

-1: (weak reject)

**Questions For Authors:**

Likely the main assertion is that the domain-independent approach can get to adequate solutions than the domain-dependent approaches?

The paper is on modeling more than on solving, no new planner is proposed?

At the end I am not happy about the general lessons to be learnt from the paper, as the case is seemingly not complex enough for being put back to the real-world?

What is added practical value of this paper? While we all know that PDDL+ can express a lot, it hardly scales.

**Reproducibility:**

5: Code and domains (whichever apply) are already publicly available

**Strengths Of The Paper:**

PDDL+ is located on the top of Fox and Long’s PDDL2.1 hierarchy with a semantic that refers to hybrid automata. Since then, there have been a number of planners proposed, mostly operating with some form of dynamic discretization, some calling alternative solvers as subroutines.

I welcome that the paper is not overloaded with technical notation, but at the same time this makes the technical level of the paper quite shallow.

The paper emphasis with its only keyword that it is an application paper.

The models were made available with a link in the text.

The writing is adequate. It introduced smoothly to UK urban traffic control (UTC). It also describes the existing SCOOT system and how PDDL+ is used not for solving but for simulating it. It is well-known that plan validator VAL is quite capable to expressive even allow for some high-level PDDL simulations.  Example PDDL+ fragments are given.

In the comparison against the SOTA the text emphasizes the problem independence of the approach, which to some extent is true, but to validate this this case they should have included more than one application area.

**Weaknesses Of The Paper:**

For me the paper looks a bit shallow and borrows its power by applying PDDL+, which perfectly fits to ICAPS, but also loods at too
large bullets for a too small problem.

I took the time look at all of the domains and instances. While the PDDL+ domain model is considerably complex with a handful of actions, processes and events, all instances seem to have at most 6 junctions (similar to the one displayed in the figure), which imho hardly is relevant in any practical route planning or simulation example. At most this can count as proof of concept. Lots of constants encode numerical quantities. As a minor the models are said to have durational inequalities, but I haven’t found duration at all. In any case, processes and events are sufficient to model time.

They compare the plans generated by FIRE against the plans historically implemented by SCOOT. Additionally, to a domain-specific planning approach designed with the EXRE model from literature. Maybe I have not looked hard enough but in the comparison I cannot find a clear-cut winner.

The conclusion state that it generated deployable traffic signal plans for a specific UTC infrastructure, which is quite a promise that I cannot see fully validated.

---

> ### Author Rebuttal · Authors · 2024-01-26
>
> We thank the reviewer for the feedback, but in the following we’d rectify some aspects of her review that may have negatively affected her overall evaluation.
>
> [VAL] We do not use VAL, but a dedicated architecture that allows to use PDDL+ to simulate traffic evolution, by leveraging on specifically-designed models (Bhatnagar et al. 22/23).
>
> [6 junctions] We do not use planning for routing vehicles, but for traffic signal optimisation. We aim to replace SCOOT -it operates in regions of 4-8 junctions. The reason they are of this size is because SCOOT deals with connected junctions where handling offsets between them is meaningful. If there are several pedestrian crossings and/or long links between junctions, they are modelled in different regions. In other words, we are dealing with real-world-size scenarios.
>
> [durations] We apologise for the confusion. As most PDDL+ planning engines do not parse the requirements section, it is commented away in our models and should be ignored. For clarity, we will remove the requirement sections from the final repo.
>
> [clear-cut winner] The proposed models use a domain-independent planning system to generate strategies that are comparable, or even better, than those generated by a domain-specific approach or those generated by the traffic control system in operation. This, in itself, is an outstanding result for domain-independent planning. On top of that, generated solutions are deployable.
>
> Q1.The main result is that the proposed models allow a domain-independent planning engine to generate solutions that can be deployed in the major UTC system of the UK.
>
> Q2. Planning and modelling are intricately connected. As demonstrated by the experiments, existing technology allows to generate good quality plans. In the future, to further extend the capabilities of the approach, we will consider the design of domain-specific techniques.
>
> Q3-4.We’d like to reiterate that this is exactly the size that deployed SCOOT operates on.
> The main advantage is that the proposed approach brings goal-directed traffic plans into the reach of traffic controllers, that’s not possible on SCOOT system or other deployed traffic control systems, which are aimed at optimising overall journey time only, and which fails to do this under unusual conditions. Further, the use of planning also allows traffic controllers to generate in advance complex strategies for major events, and to compare different strategies according to expected traffic demands.

---

### Meta-Review · Area_Chair_CpQD · 2024-02-06

**Recommendation:** Accept (Poster)
**Confidence:** 5

**Metareview:**

On balance I recommend accepting this paper. While there are concerns raised about the potential for impact due to the scale of the roadmap, it is nonetheless of a reasonable size for a PDDL+ benchmark; and having good and persuasive PDDL+ benchmarks is important to help guide and encourage the development of practically useful techniques for PDDL+ planning.

**Ethical Considerations:**

(1) Not Applicable: The paper does not have any ethical considerations to address